# Quantifying societal emotional resilience to natural disasters from geo-located social media content

**Krishna Bathina**[1]*, **Marijn ten Thij**[1,2,3], **Johan Bollen**[1]

**1** School of Informatics, Computing, and Engineering, Indiana University, Bloomington, IN, United States of America, **2** Delft Institute of Applied Mathematics, Delft University of Technology, Delft, Netherlands, **3** Department of Data Science and Knowledge Engineering, Maastricht University, Maastricht, Netherlands

* bathina@indiana.edu

**Data Availability Statement:** All relevant data are within the paper and its Supporting information files.

**Funding:** Johan Bollen thanks the National Science Foundation (SMA-SBE: 1636636, https://www.nsf.

## Abstract

Natural disasters can have devastating and long-lasting effects on a community's emotional well-being. These effects may be distributed unequally, affecting some communities more profoundly and possibly over longer time periods than others. Here, we analyze the effects of four major US hurricanes, namely, Irma, Harvey, Florence, and Dorian on the emotional well-being of the affected communities and regions. We show that a community's emotional response to a hurricane event can be measured from the content of social media that its population posted before, during, and after the hurricane. For each hurricane making landfall in the US, we observe a significant decrease in sentiment in the affected areas before and during the hurricane followed by a rapid return to pre-hurricane baseline, often within 1-2 weeks. However, some communities exhibit markedly different rates of decline and return to previous equilibrium levels. This points towards the possibility of measuring the emotional resilience of communities from the dynamics of their online emotional response.

## Introduction

Hurricanes are devastating to their affected communities. The US alone has had over 30 hurricanes in the last 20 years, whose aggregated effects involving destroyed infrastructure and loss of life are difficult to estimate. This is all the more true for a hurricane's social and psychological effects on well-being. Some communities suffer long lasting and immeasurable damage to their social fabric, even after extensive government and community intervention, and reconstruction of infrastructure.

Much of the attention surrounding hurricanes has focused on physical issues and logistics, such as preparing and protecting infrastructure, and the prevention of loss of life, but the social and psychological consequences can be equally relevant. For example, after Hurricane Katrina, New Orleans lost 53% of its population, leaving entire wardens depopulated [1]. In addition, the disaster had a profound effect on the well-being of the New Orleans community which was not fully captured by subsequent attempts to assess the economic damage [2]. Whereas some

gov/) and the Economic Development Administration (EDA/ED17HDQ3120040, https://www.eda.gov/), Wageningen University (the Netherlands), the Vice Provost for Research at Indiana University, The Center for Mental Health at the University of Amsterdam, and the ISI Foundation (Turin, Italy) for support. The funders had no role in study design, data collection and analysis, decision to publish or preparation of the manuscript.

**Competing interests:** The authors have declared that no competing interests exist.

areas did make a partial recovery and benefited from the construction of new public service systems, others struggled to return to their pre-hurricane situation.

The mental and psychological effects of natural disasters may play an important role in the recovery of a community after a natural disaster. This important facet of recovery is difficult to quantify other than painstakingly conducting population surveys and interviews at regular intervals [3]. Unfortunately, this approach presents considerable logistic challenges with respect to scale, costs, and accuracy.

Here, we look to social media as a potential source of complementary real-time proxy data to model changes in geo-located, regional well-being. We measure the collective emotional response and subsequent recovery of geographically defined communities from large scale Twitter data, in the period before, during, and after a hurricane to estimate its resilience to external shocks at high temporal resolution.

We do so from the perspective that resilience is a characteristic of systems that can resist external shocks, either by returning to their previous equilibrium or by developing an adaptive response to the external driver. In this work, we characterize the emotional resilience of a community as its "time to recovery", namely its ability to resist the negative effects of a natural disaster and return to baseline sentiment afterwards. By measuring the dynamics of changing community sentiment from social media before, during, and after a natural disaster, we attempt to model the characteristics of its emotional resilience.

Our work builds upon the foundation set by previous research about social media and well-being. Jaidka et al. [4] used 1.53 billion geo-located tweets in the United States and showed that word-level methods of sentiment analysis provide good estimates for county-level subjective well-being when compared to Gallup survey measures. Fan et al. [5] used Twitter timelines from 74,487 users and analyzed tweets before and after an explicit report of a positive or negative emotion showing that reports of a positive emotion were preceded by a sharp increase in sentiment followed by a shallow drop to normal levels while reports of negative emotions showed the opposite pattern: a slow buildup of negative sentient followed by a sharp drop and a slow return to baseline levels. Valdez et al. [6] used an analogous method to study the effects of the COVID-19 pandemic on sentiment in the US from January to April.

Our work also builds upon previous research in social media analysis and community response. Bollen et al. performed sentiment analysis on Tweets from 2008 and found that changes in sentiment correlated with major and notable events that year [7]. Bollen et al. and Yang et al. both studied Twitter data and found sentiment to be correlated to stock market returns [8, 9]. Recently, there have also been many papers studying Twitter sentiment during the COVID-19 pandemic, specifically focusing on the the effects of lockdowns and restrictions [10–13].

Much work has been done surrounding the use of online social media to study hurricanes. Arnold et al. showed that the volume of tweets and hashtags increase as a function of the intensity of hurricanes [14]. Stark et al. showed that the frequency of tweets related to Hurricane Irma peaked as the hurricane hit the region and that the level of concern varied by both region and gender [15]. This was further developed by Neppalli et al., who showed that the sentiment also varies by distance to the disaster [16, 17]. Zou et al. used tweets to highlight the disparity of social media usage during Hurricane Harvey [18]. Communities with better social and geographical conditions were shown to send more disaster related tweets.

Here we build upon these works by examining the emotional response of geographical communities to hurricanes as an instance of natural disasters that affect community well-being. We posit that hurricanes form a good case-study to assess community recovery and resilience for the following reasons:

**Unpredictability:** Hurricanes can vary unpredictably in terms of their specific magnitude and geographic scope, making it difficult for communities to anticipate their social and economic effects.

**Synchronization:** Hurricanes affect all residents of a community in a geographic location at a discrete point in time, allowing analysis of sentiment dynamics centered on a particular point in time.

In this work, hurricanes serve as natural experiments of how natural disasters may drive changes in the well-being of disparate communities. In fact, the well-being of entire communities can be observed before, during, and after hurricanes in order to study and model the community's resilience to significant natural disasters. To this end, we examine the emotional response of four geographically distinct communities on social media over three Atlantic hurricane seasons in the context of 4 hurricanes: Irma (Florida), Harvey (Texas), Florence (Carolinas), and Dorian (Carolinas, Florida, and Alabama). We downloaded geo-located tweets from the Hurricane Formation (tropical cyclogenesis) to Hurricane Dissipation (tropical cyclolysis) for each hurricane. We define "US Landfall" as the day when the hurricane makes landfall in the region of interest. We also define the following time periods:

**Before:** Hurricane formation to 5 days before US Landfall

**During:** 5 days before US Landfall to 5 days after US Landfall

**After:** 5 day after US Landfall to Hurricane Dissipation

As shown in Fig 1, after recording the hurricane location and time period, we use the Observatory on Social Media (OSoMe) [19] and the Twitter API to download tweets that were posted in the affected region. We then extract the valence of the resulting sets of geo-located tweets using a sentiment analysis tool to study the pattern of changing Twitter sentiment during the hurricane. Next, we bootstrapped the calculated sentiment and then built exponential fits. Finally, to discover terms that are characteristic to the changes in sentiment, we study the frequency of VADER lexicon terms **before**, **during**, and **after** the hurricane landfall.

We analyzed changes in Twitter sentiment in the regions affected by the set of selected hurricanes, including 2 control regions that were not directly affected by a hurricane or where the specific hurricane did not make landfall. We modeled the trend of changing sentiments before,

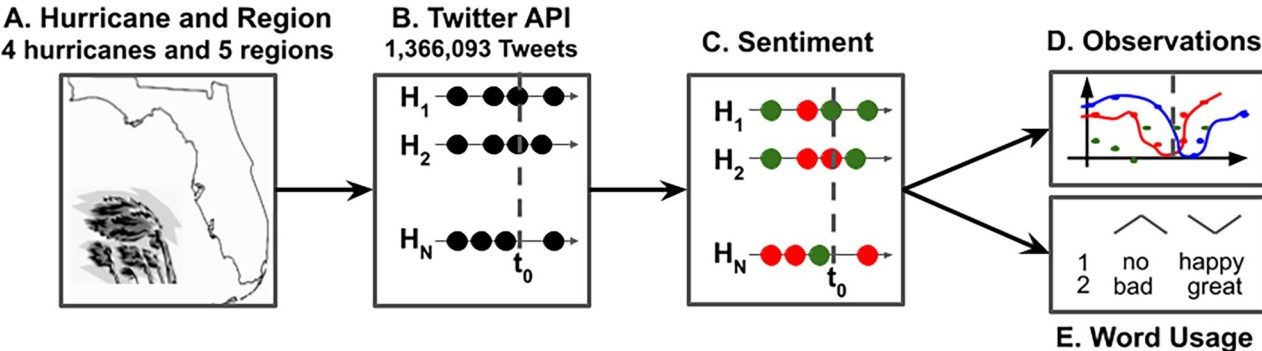

**Fig 1. An outline of our methodology.** A) We retrieve bounding box coordinates of the US Landfall of 4 major hurricanes in 5 regions. B) We then download the geo-located tweets from the affected region during the hurricane using the Twitter API. C) Using sentiment analysis tools, we quantify the valence of the tweets for. D) We bootstrap the sentiment and then build exponential fits. E) Finally, we look at the change in frequency of VADER lexicon terms **before**, **during**, and **after** the hurricane landfall.

during, and after each respective hurricane. Our results indicate that in all cases, Twitter sentiment in each affected region slowly decreases as the hurricane approaches indicating widespread anticipation in the respective populations, a large drop right after US Landfall reflecting the effects of the hurricane making landfall, and a rapid returns to previous levels over the course of about 1–2 weeks. Our results indicate that Twitter sentiment does reflect changes in population subjective well-being as a result of a discrete negative event and that the rate of change over time reflects community-specific response rates which may be indicative of emotional community resilience.

## Background on chosen hurricanes

Hurricane Irma, originating from Cape Verde, was the first Category 5 hurricane in 2017. In September, the hurricane formed in the Caribbeans and hit Puerto Rico and Florida. Because of advanced warning systems, many of the Caribbean islands were prepared for evacuations. Florida declared a state-of-emergency causing the evacuation of 6.5 million Floridians and a reroute of all inbound flights. Approximately 73% of the state was left without power and 65,000 structures were destroyed [20, 21]. The damage was estimated to be at 50 billion dollars with 84 deaths [22]. Hurricane Harvey was a Category 4 hurricane that formed near Barbados in August 2017. It moved northwest towards Mexico and hit lower east Texas. It tied Hurricane Katrina for causing the most economic damage, evaluated at 125 billion dollars [23]. An early warning storm watch, multiple state-of-emergencies, and mandatory evacuations were declared to minimize loss but over 100,000 homes and vehicles were still destroyed [24]. There were electricity outages across the path of the hurricane and 103 deaths in Texas alone.

Hurricane Florence, originating from Cape Verde, hit the Carolinas in September 2018. It was the first major hurricane of the 2018 season causing a state-of-emergency, an overnight curfew, and mandatory evacuations as rains reached a record of 34 inches in North Carolina. The deluge caused large sections of the interstate to close down and many homes were completely flooded. There were over 50 confirmed deaths and estimated damages of over 24 billion dollars [25, 26].

Hurricane Dorian formed near Senegal in August 2019 [27]. The hurricane moved west and missed a majority of the Caribbean islands. Unfortunately, it was one of the most destructive storms to hit the Bahamas. The slow moving nature of the hurricane caused immense damage, leveling a majority of islands it came in contact with. The hurricane then moved north of Florida and landed in the Carolinas on September 5th with winds up to 90 mph [28]. Both Florida and the Carolinas were under a state-of-emergency due to flooding and high winds [29, 30]. Both areas had 3 deaths total but the Carolinas had over 160,000 buildings with no power [31–33].

Before the hurricane hit, the president of the United States Donald Trump stated that "Alabama, will most likely be hit (much) harder than anticipated" on his personal Twitter account and released a press release with the same information on the White House official Twitter account. This was refuted by the National Weather Service and the hurricane did not affect Alabama, but the official statement by President Trump was never refuted [33–35].

Fig 2 shows the trajectories of each hurricane as well as the geographic bounding boxes corresponding to the affected areas from which tweets were sampled. Note that we analyze Hurricane Dorian's effect on 3 distinct regions as a control: (1) Florida which it skimmed but where it did not make landfall, (2) Alabama which it did not affect at all, but where, due to misinformation on social media, some of the public may have believed they could be affected, and (3) the Carolinas where the hurricane did make landfall.

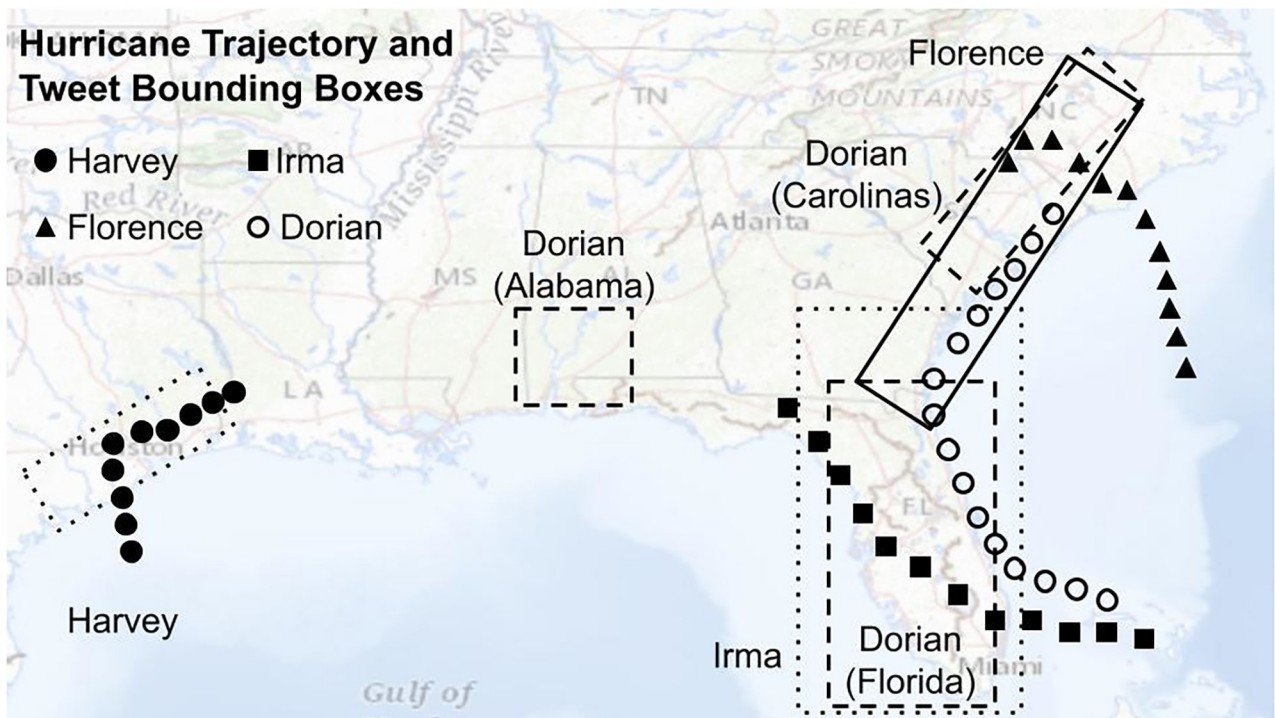

**Fig 2. The trajectories of the hurricanes and the bounding box of the Tweet Samples.** The east coast of the United States gets hit with many yearly hurricanes because the warmth of the water alongside the coast provides thermal energy to incoming hurricanes. Hurricane Harvey came up from the south and hit south-east Texas, especially the Houston metropolitan area. Hurricane Irma also came up from the south but landed in Florida. Hurricane Florence came across the east coast and landed in the Carolinas. Hurricane Dorian skimmed Florida but also made landfall in the Carolinas. Alabama was not affected by Dorian even though an official memo from President Trump claimed they would get hit.

## Data and methods

### Data

We use social media for this analysis because of its pivotal position in public discourse. According to the Pew Research Center, 84% of US adults aged 18–29 used social media in 2021. Among those older than 65 years, 45% report that they are social media users. Even though Facebook has the most users (69% of US adults), we specifically use Twitter due to its microblogging nature; it is most commonly used to discuss daily routine and reporting news [36] while still representing 23% of all social media users. This is particularly important for our analysis as we are interested in people's evolving reports of their personal state in the context of a hurricane.

We collected geo-located Twitter data that was posted between 2016 to 2019 from (OSoMe), which provides a 10% random sample of all daily tweets posted worldwide. Here we only retained tweets whose geo-location data matched the bounding boxes corresponding to the respective hurricane-affected regions shown in Fig 2. The S1 Table shows a summary of the raw data. In our subsequent analyses, we excluded all retweets and non-English tweets, as identified by the Twitter API that provides this information. The number of analyzed Tweets for each hurricane and region are shown in Table 1. A list of Tweet IDs for tweets used in this analysis can be found at https://github.com/kbathina/resilience-to-natural-disasters. All data

**Table 1. Hurricane data.**

| Hurricane | Location | Number of Tweets |
|---|---|---|
| Irma (2017) | Florida | 823537 |
| Harvey (2017) | Houston | 511229 |
| Florence (2018) | Carolinas | 500332 |
| Dorian (2019) | Carolinas | 7921 |
| Dorian (2019) | Florida | 20646 |
| Dorian (2019) | Alabama | 2760 |

The name, year, and location of each hurricane followed by the raw number of tweets downloaded from OSoMe.

from OSoMe and the Twitter API were collected in accordance with their respective terms and conditions.

Fig 3 shows the weekly and daily frequency of filtered tweets. The regions affected by hurricanes Irma, Harvey, and Dorian in the Carolinas show an increase in activity around their US Landfalls, indicating that user activity increased during the hurricane and that Twitter may thus be suitable to gauge geo-located hurricane-related communications.

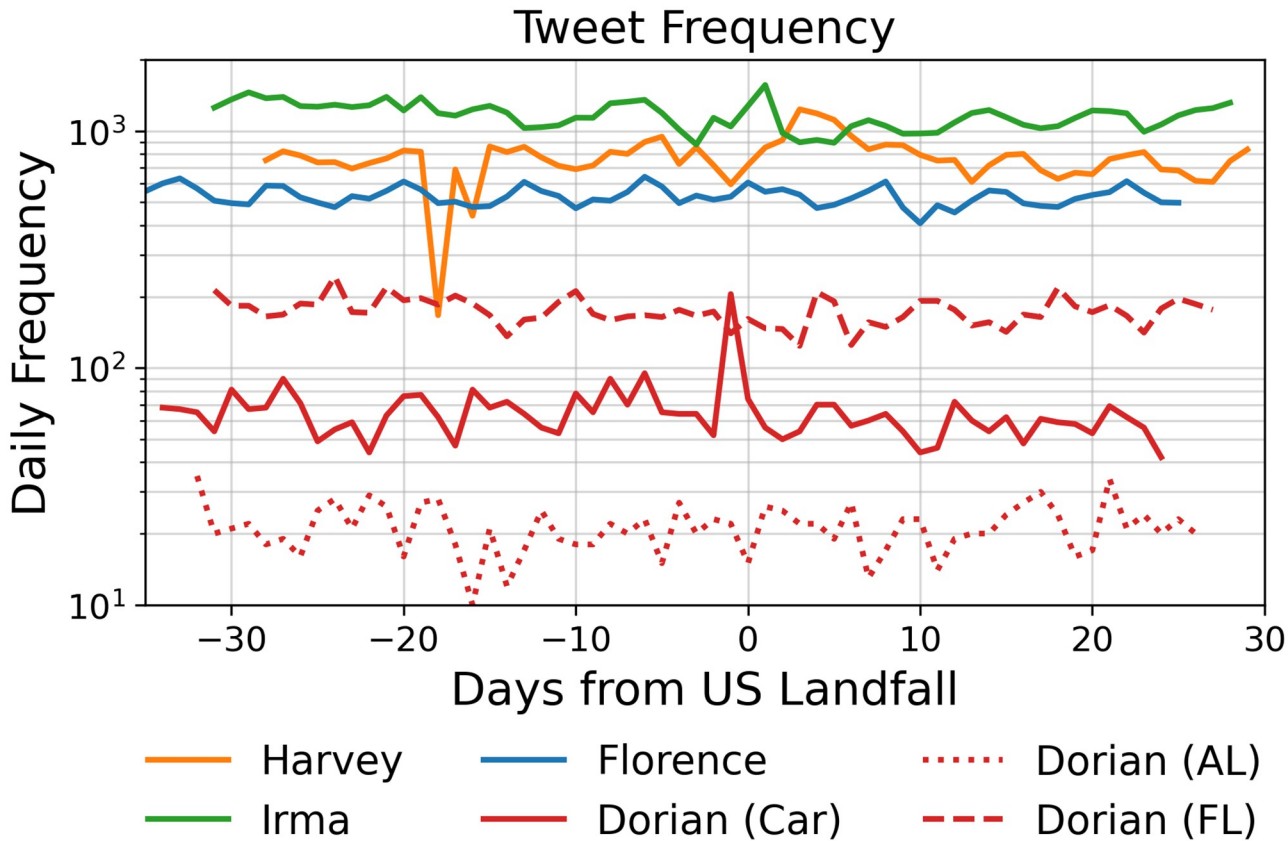

**Fig 3. Daily tweet frequencies for Hurricanes Irma, Harvey, Florence, and Dorian.** All dates are rescaled to the number of days since the US Landfall the area. In the case of Hurricane Dorian in Alabama, US Landfall is when President Trump predicted the hurricane would hit.

## Sentiment analysis

Sentiment analysis refers to a collection of supervised or unsupervised natural language processing technique that maps the content of a text to a rating or classification of its sentiment or emotions.

To assess the sensitivity of our analysis to the choice of sentiment analysis tool, we subject each tweet to two different sentiment analysis tools. First, we use the **Valence Aware Dictionary for sEntiment Reasoner (VADER)**, a tool highly suitable for Twitter content [37] using a large lexicon that was developed through crowd-sourcing sentiment rating. VADER matches the terms in a tweet to an extensive lexicon of 7,517 terms, applying a set of grammatical rules that acknowledge common use of punctuation, capitalization, modifiers, and negation that can affect a tweet's sentiment, e.g. "I am extremely NOT pleased!!!!". VADER returns a Tweet's Valence rating on a scale from -1 (negative valence) to +1 (positive valence). Because VADER does not natively differentiate between the absence of a match and neutral (zero) sentiment, we modified VADER to output 0 for neutral sentiment and "NULL" for no matching words.

Second, we compare VADER sentiment values for each tweet with those generated by the **Linguistic Inquiry and Word Count (LIWC)** [38] tool. Using a predefined dictionary, LIWC returns a positive (LWPR) and negative (LWNR) word ratio—i.e., the number of positive or negative words divided by the total number of words in the tweets. Besides calculating different values, LIWC and VADER use different sentiment lexicons. In an earlier large-scale bench-marking test for a random sample of tweets, VADER was shown to outperform LIWC in both macro-F1 score (57.89 vs, 35.95) and coverage (82.20% vs. 61.82%) respectively [39].

Hurricanes invoke bleak imagery and are generally spoken about in a negative context. Any significant changes in sentiment could therefore be an artifact of Twitter language shifting towards the topic of hurricanes (i.e., greater prevalence of hurricane-related words) and sentiment analysis reflecting their inherently negative sentiment. However, we checked to see if a set of hurricane-related words from our analysis, such as "hurricane", "tornado", "storm", "rain", "deluge", and "flood", exist in the VADER or LIWC dictionary. Both dictionaries did not contain any of these words, so our sentiment results are likely not skewed by the presence of topical hurricane-related words in the Twitter data set.

## VADER lexicon usage

To characterize the emotional response of a region to a hurricane, we look for the VADER lexicon words, i.e. words with a valence loading, that are most characteristic of changes in sentiment during a hurricane. In particular, we look for VADER lexicon terms with the largest change in frequency, positive or negative, **during** the hurricane relative to their frequency of usage **before** and **after**.

We first grouped the tweets of each region into three time periods (**before**, **during**, and **after** the hurricane) as defined in the Introduction. We then tokenized each tweet and removed words that are not included in the VADER lexicon because we are only interested in words that affect our observation of tweet valence. Next, we calculated the TF-IDF of each word during each time period, expressing how specific the word is to the time period. We then z-score normalize the TF-IDF values of each word using the mean and standard deviation across the 3 time periods, resulting in a 3×1 vector of normalized TF-IDF scores for each word. As a result, each word's vector contains a **before**, **during**, and **after** z-score normalized TF-IDF score that expresses the degree to which the TF-IDF value of the term for a period

deviates from the 2 other periods, i.e. how well does the term characterize a particular time period relative to the 2 other periods.

Using the words as labels and the 3×1 TF-IDF vectors as features, we ran a k-means clustering with $k = 2$ clusters. We observe that for each region the same two clusters of patterns emerge; a concave shape (down, up, down) representing an increase in the z-score normalized TF-IDF scores **during** the hurricane and a convex shape (up, down, up) representing a decrease in the normalized TF-IDF scores **during** the hurricane.

For each region and hurricane, we have a set of TF-IDF vectors of words grouped into these two clusters (up during the hurricane vs. down). In order to determine if there was a change in the importance of VADER words **during** the hurricane, we compare the rank order of the words between time periods using the z-score normalized TF-IDF value. One such comparison is the Kendall rank correlation coefficient ($\tau$), which is a measure of correlation, or similarity, between two rankings. A high correlation corresponds to a nearly similar ranking order while a low correlation represents a dissimilar ranking.

## Bootstrapping and null-model

The volume of tweets and the Twitter user sample changes with time. These differences can lead to variance and estimation errors between time windows, complicating comparisons of sentiment over time. Therefore we bootstrap our estimates of sentiment values to obtain 95% confidence intervals of the mean sentiment at a given time as follows. Given a timeline of tweets split into $t_k$ time units with $N_k$ tweets each, $\mathcal{O}(t_k)$ is the set of observed sentiment z-scores at time $t_k$ for tweets that have a calculated sentiment (see subsection "Data"). To calculate if a day $k$ is statistically significantly high or low, we first bootstrap by calculating the mean sentiment of a random sample with replacement of $\mathcal{O}(d_k)$ of size $N_k$. This bootstrapping process is repeated $B = 10,000$ times for every day, $\bar{\mathcal{O}}^*_1(d_k), \bar{\mathcal{O}}^*_2(d_k), ..., \bar{\mathcal{O}}^*_B(d_k)$.

To compare observed sentiment against a random sample of tweets, we also build a daily null-model, $\mathcal{N}(d_k)$, by only randomly sampling (with replacement) from tweets in $\mathcal{O}(d_{k-28})$ to $\mathcal{O}(d_{k-1})$ and at the same day of the week as $d_k$. The null-model is also bootstrapped $B$ times per day, $\bar{\mathcal{N}}^*_1(d_k), \bar{\mathcal{N}}^*_2(d_k), ..., \bar{\mathcal{N}}^*_B(d_k)$.

Since the null-model indicates the expected sentiment for a random sample of tweets, we compare observed sentiment values to this null-model to gauge their significance. We do this by first calculating the 2.5th, 50th, and 97.5th percentile of the bootstrapped values of each day for both the observed data and the null model. Days in which the 97.5th percentile of the observed bootstrapped sentiment is less than the 2.5th percentile of the null model bootstrap indicates a statistically significant decrease in sentiment. Similarly, days in which the 2.5th percentile of the observed bootstrapped sentiment is higher than the 97.5th percentile of the null model bootstrap indicates a statistically significant increase in sentiment.

## Valence fits

To estimate the rate of sentiment decay and recovery surrounding each hurricane, we fit several models to each region's sentiment over time. We grouped Twitter sentiment (each tweet has one valence value) in 12 hour windows and then calculated the mean tweet valence value for each window. We then fit 2 curves; one descending curve ending at the minimum sentiment and one ascending curve starting from the minimum sentiment onwards [5]. The sum of squared errors (SSE) for each model are shown in the S2 Table. The exponential fit in the form $E = ae^{bt} + c$ where $t$ is the time in days, $E$ is the expected sentiment, and $e$ is the natural exponential had the lowest SSE.

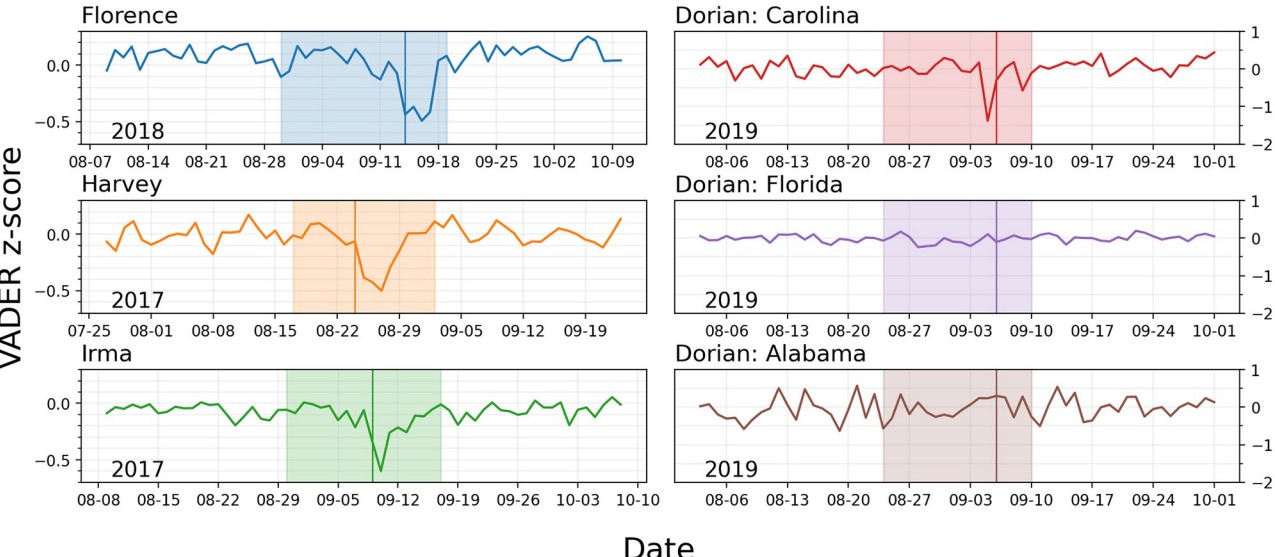

**Fig 4. Z-score normalized sentiment of tweets in each region.** For each affected region, the sentiment was the lowest during the hurricane. In the case of Hurricane Dorian, only the Carolina showed a large drop. Alabama and Florida, on the other hand, showed very little changes in sentiment throughout the whole time period.

## Results

We first look at the sentiment and significant changes over the period under consideration (2017 to 2019) when Hurricanes Harvey, Irma, Florence, and Dorian affected the US. Fig 4 shows the z-score normalized VADER sentiment (subtracting the mean and divided by the standard deviation of the time series for that period) for each region. In each instance of US landfall, the lowest VADER sentiment (z-scores) was observed during the hurricane. Note that in the case of Hurricane Dorian, only the Carolinas showed a decrease in VADER sentiment as the only region to be directly affected, contrary to Alabama and Florida that served as controls in this analysis. These observations support the notion that hurricanes are associated with changes in collective sentiment in the affected regions and that these changes are contingent on the hurricane making landfall in the given region.

To determine whether statistically significant changes in sentiment occurred at a given point in time relative to when the hurricane made landfall, in Fig 5 we plot VADER sentiment changes relative to a null-model of randomly selected tweets (see Materials and Methods) for each daily time window. The null-model we use compensates for seasonal effects in our data by taking a random sample of tweets from the period shortly before the time point itself. This comparison reveals statistically significant daily sentiment changes relative to when the respective hurricane made US landfall. For Hurricanes Florence, Harvey, and Irma, we observe a large drop in sentiment that occurred within the first 4 days of landfall. Hurricane Dorian in the Carolinas, on the other hand, shows a large drop only one day before US Landfall. In each of these regions, VADER sentiment decreased significantly while the LWNR increased significantly (S4 Fig). The LWPR (S3 Fig) showed no pattern of significant change in the affected areas and the control groups.

After the sustained negative sentiment (about 8 days after landfall), the VADER sentiment increases significantly, whereas the LIWC negative word ratio (LWNR) decreases significantly, indicating a comparably more positive sentiment. We observed a significant increase in the LIWC positive word ratio (LWPR) for Hurricane Irma only. One may speculate this indicates

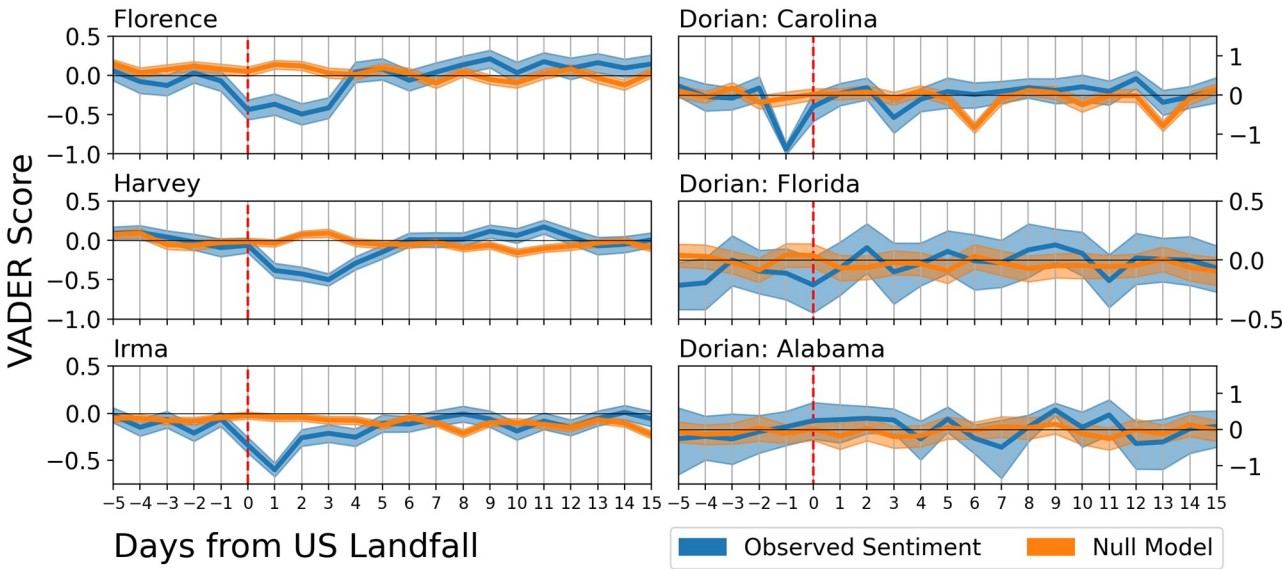

**Fig 5. Bootstrapped VADER sentiment around each hurricane and location.** For each day, we calculated the 2.5th, 50th, and 97.5th percentile of the bootstrapped sentiments in both the original data and the null model. The bold line is the median of the observed data and the null model while the shaded areas represent the 95% confidence interval. Any day in which the confidence intervals does not intersect represents a statistically significant change in sentiment. In all locations with landfall, the overall sentiment was lower when compared to the previous month. This was followed by 2–3 days of a sustained depressed sentiment. After about 8 days, the sentiment was statistically higher due to the emotional resilience of the community.

a potentially greater emotional resilience of this community, assuming that the ability to return to previous levels after an exogenous shock indeed corresponds to community resilience. As expected, there were few significant changes in Florida and Alabama before, during, and after Hurricane Dorian since physical damage was limited in those states.

We also calculated the magnitude of sentiment change using the daily, observed z-score medians. The fits and sum of squared errors (SSE) for each hurricane using VADER are shown in Table 2 and the plots are shown in Fig 6.

Fig 6 and in Table 2, the Carolinas were the only region to have the minimum sentiment one day before the hurricane made landfall, as opposed to the other regions, in which the lowest sentiment was one to four days after. Fig 4 also shows that the Carolinas had the largest decrease in sentiment; almost double the other affected regions. However, the Carolinas also bounced back to normal in almost 2–3 days, indicative of high emotional resilience. Krause et al. found it took 16 months for an individual to mentally recuperate from the stressors of

**Table 2. Exponential fits of VADER sentiment for Hurricanes Irma, Harvey, and Florence.**

| Hurricane | Descending | | | Ascending | | | | |
|---|---|---|---|---|---|---|---|---|
| | a | b | c | a | b | c | SSE | Half-life (Days) |
| Florence | -0.34 | 0.26 | 0.10 | -12.99 | -1. | 0.11 | 1.45 | 0.5 |
| Harvey | -0.11 | 0.62 | -0.01 | -7.22 | -0.77 | -0. | 1.28 | 0.9 |
| Irma | -0.18 | 0.80 | -0.06 | -1.3 | -0.64 | -0.06 | 0.72 | 1.3 |
| Alabama | / | / | / | / | / | / | / | |
| Florida | / | / | / | / | / | / | / | |
| Carolina | -2.68 | 1. | 0.06 | -0.66 | -0.95 | 0.07 | 10.38 | 0.7 |

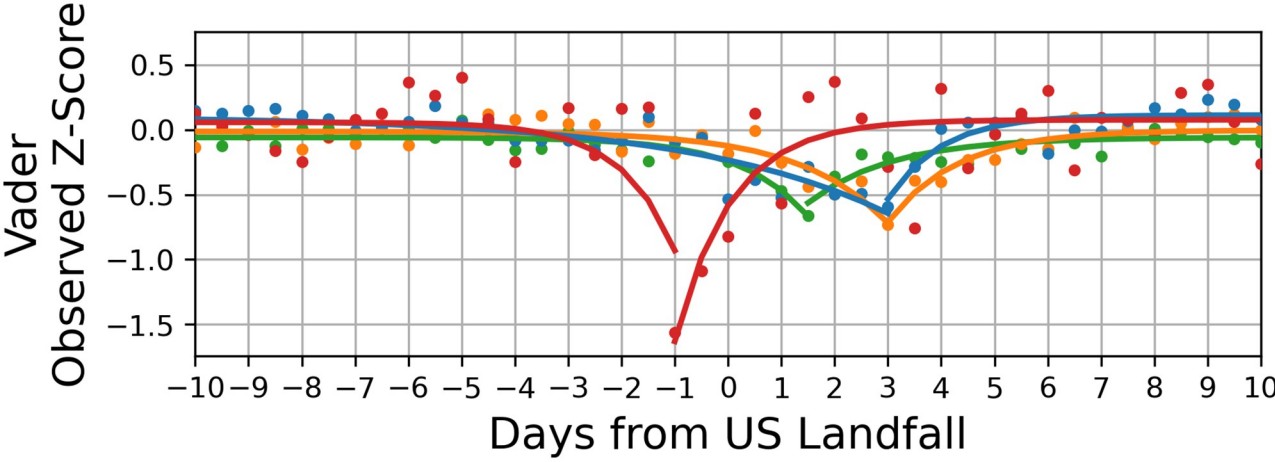

**Fig 6. Exponential fits of VADER sentiment and change in sentiment.** The first plot shows every 12 hour grouped VADER data. With this, we fit two exponential function to each affected area using. The exponential form is $E = ae^{bt} + c$ where $t$ is the time in days, $E$ is the expected data, and $e$ is the natural exponential.

natural disasters [3]. One theory to explain the differences in recovery time with our results is that two different types of thinking are being measured [40]. The quick resilience can be explained by an immediate and innate relief (Type 1 thinking) immediately after the primary disaster of the hurricane. The longer resilience can be explained by a logical and more analytical thinking (Type 2 thinking) involved in fully realizing the long- and short term effects of the hurricane. The pattern for the other affected regions were very similar, except with a much smaller impact. We don't show fits for Florida and Alabama as there was no significant sentiment change near the hurricane event, even though Dorian skimmed by the Florida coast and many may have been worried about the possibility of it making landfall in that area.

Finally, we looked at all words with sentiment loadings, i.e. words that exist in the VADER lexicon, that were used in tweets posted from the respective communities **during** the hurricane event, and compared them to words **before** and **after** the hurricane. Fig 7 shows the Kendall rank correlation coefficient between time periods in each cluster. There is a high rank correlation between VADER words **before** and **after** the hurricane, thus showing a very little difference in rank order, i.e. the importance of the valence words are similar in the two time period before and after the hurricane. The rank correlation when comparing with words **during** the hurricane are 8–9x times lower, indicating a large difference between the rankings.

The S1 and S2 Figs show the top 5 valence words in tweets with the largest z-score normalized TF-IDF score in each cluster and time period. **Before** and **after** the hurricane, the most important words were the same in each region, as shown in S1 Fig. For example, Hurricane Florence had the same exact word list in the convex cluster **before** and **after**. **During** the hurricane, on the other hand, "warning" became the most important word. The concave cluster has the same group of words in all of time period. The decrease in TF-IDF scores of the top 5 words **during** the hurricane can be attributed to an increase in the use of the word "warning" from the convex cluster. A similar pattern can be seen in Hurricanes Harvey and Irma as well; the concave clusters have the exact same order of importance across all time periods and the decrease in importance **during** the hurricane can be attributed to an increase in frequency of words such as "help" and "safe" in the convex cluster. In Hurricane Dorian, Florida and Georgia showed very little difference in the top 5 most important words. In the Carolinas, on the

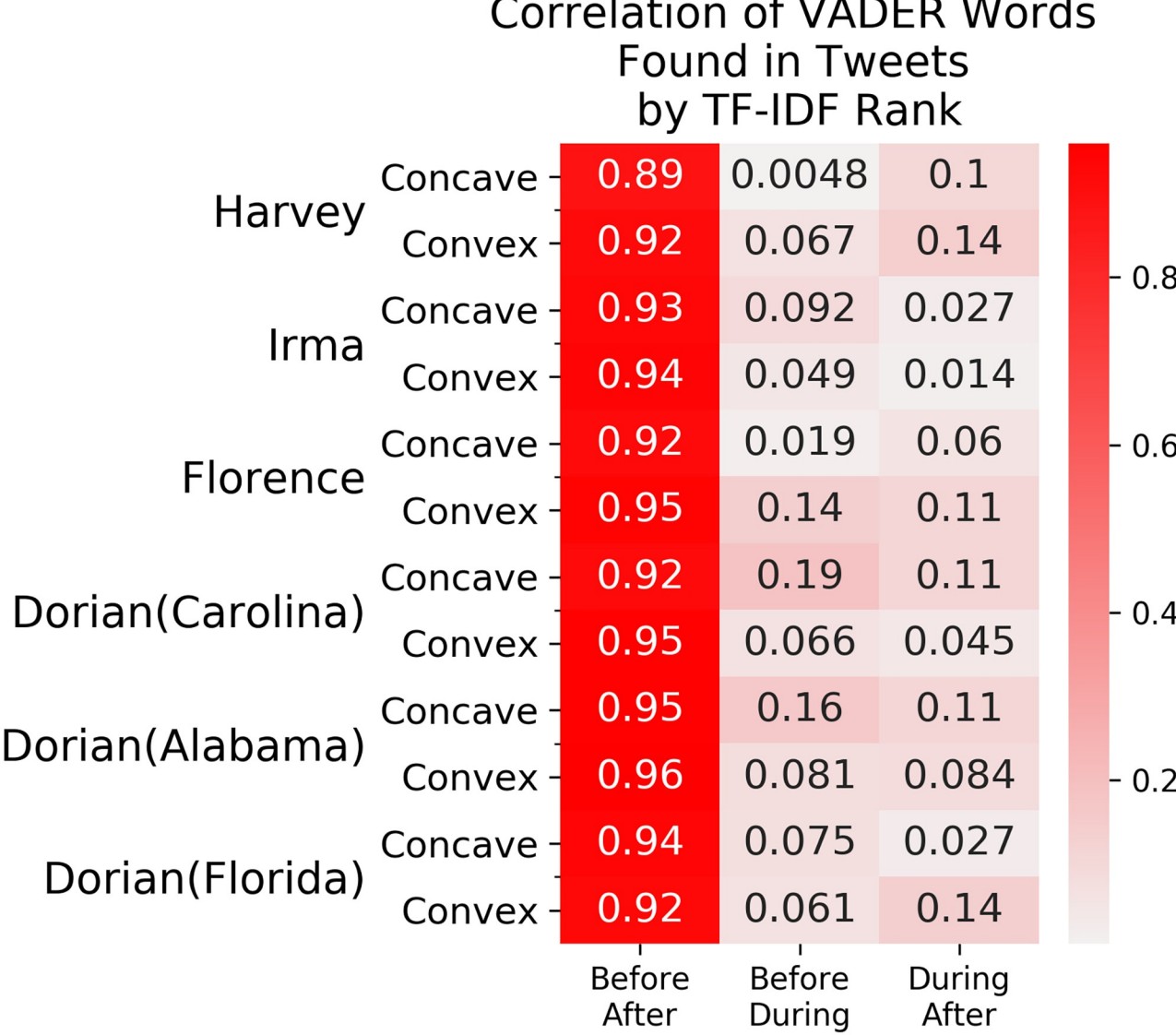

**Fig 7. Rank order correlation of VADER words by TF-IDF.** The Kendall rank correlation coefficient is a measure of correlation between two orderings. A high correlation between orderings of TF-IDF values of VADER terms from two time periods indicates a similar ranking of the VADER terms between the two time periods. The correlation between the **before** and **after** orderings are almost 1, indicating a near perfect similarity in the word rankings. On the other hand, all of the calculated correlations involving the **during** orderings are almost 8–9x less, indicating that the hurricane had an effect on the usage of VADER terms in tweets.

other hand, "warning" and "safe" increased in importance **during** the hurricane while words such as "great", "love", and "happy" decreased.

## Discussion

Using geo-located social media, we show a statistically significant decrease in community sentiment during four hurricanes, even though a set of common hurricane related words had no effect on our analysis. Using exponential fits, we also show a significant emotional resilience about 1–2 weeks after the US Landfall, bringing the sentiment back to its baseline value. We

also show an increase in the importance of "danger" and "safety" related words that are not directly hurricane-related.

Our data sources and analysis has many advantages over traditional survey and interview methodology. Surveys can be impractical for live monitoring public sentiment, in particular during a crisis situation such as a hurricane making landfall. They can furthermore be biased by factors related to social expectations and social conformity bias. Pluralities of the US population are now active on social media platforms, yielding data that is recorded at high temporal resolution [41], allowing systems to monitor changes in public sentiment in approximately real-time during major events and even natural disasters.

There are some limitations to this approach that should be addressed. First, we use aggregate Twitter sentiment a a proxy of evolving community sentiment because tweets tend to be unprompted and real-time. The Twitter population is not a representative, but rather self-selected sample of the United States population [42] and a sub-sample that is active on this particular social media platform (Twitter vs Facebook, Reddit, etc), complicating efforts to generalize from this analysis. Second, hurricanes and other extreme weather can lead to power outages, sometimes lasting for days. Consequently, social media and general internet usage in some users can decrease during those times, also leading to a reliance on more traditional sources of information such as TV and radio [43, 44]. However, we believe that internet and social media usage during natural disasters is more nuanced when taking into account the "Digital Divide" [45]. As mentioned above, the Twitter population is not a representative sample of the world. For example, young adults tend to use social media more often than older adults [41]. Kent and Capello showed that during the Horsethief Canyon Fire in September 2012, younger people had disproportionately more fire related Twitter posts [46]. Xiao et al. showed that during Hurricane Sandy, the number of related tweets increased in mildly damaged places and decreased in more affected areas [47]. They also found that young, male, or more educated populations are more likely to have more tweets. When looking at income disparity and tweet frequency, they found a reversed U-shaped pattern and hypothesized that an increase in wealth can indicate better access to technology and information. But, after a certain point, the extremely wealthy are less motivated to use social media, potentially because they are less affected by the natural disaster. Furthermore, Wang et al. found similar results, showing that communities with higher social vulnerabilities were underrepresented even before Hurricane Sandy made landfall, indicating less social media engagement and the potential to be left behind during disaster recovery due to low visibility [48]. We acknowledge the Digital Divide as a limitation in our analysis as we are not able to account for these population differences. Finally, in this work, we specifically focus on major hurricanes that made landfall at different locations in the US, affecting different communities. However, our results may not generalize to other natural disasters such as floods or forest fires which may have specific effects on public sentiment and community resilience.

These limitations suggest avenues for further research. Platforms such Facebook, Reddit, and Weibo can provide detailed real-time data on public sentiment that would allow generalization beyond specific social media platforms and their communities. Combining such data with attention-level data, e.g. Google searches, and surveys has been shown to lead to greater accuracy of observations [49], and may similarly support the analysis of community resilience with respect to the short and long term effects of hurricanes. Natural disasters occur throughout the world, where many of Twitter's users reside. Multi-lingual sentiment analyses tool could therefore be useful for studying the effects of natural disaster, including but not limited to hurricanes, from a global perspective. This is particularly relevant since countries can differ significantly with respect to demographics, population density, government intervention strategies, and infrastructure. Finally, a natural extension of our work is to include more natural

disasters, such as tornadoes, floods, and earthquakes. The differences in the accuracy of forecasting between these natural disasters could lead to interesting insights of how uncertainty affects the public's emotional response to the particular disaster as well as their collective resilience.

## Supporting information

**S1 Table. Data parameters from Twitter.** The first two columns are the name, year, and location of the hurricane. The next two are the bounding box from which the geo-located tweets were sampled from. The following three date columns are the dates from which the tweets were sampled and the US Landfall respectively. In the case of Alabama, the US Landfall represents when the official announcement was made by the White House.
(PDF)

**S2 Table. Equation Fits of VADER scores.** We tested 4 different models in order to find the best fits. In all cases except Dorian in the Carolinas, the exponential fit had the lowest sum of square errors (SSE).
(PDF)

**S1 Fig. Top 5 VADER words by normalized TF-IDF scores before, during, and after each hurricane.** The first cluster showed more caution related words with a higher TF-IDF **during** the hurricane than **before** or **after**, such as "warning', "safe", and "help". In contrast, the second cluster had very similar words across all time periods. The drop in TF-IDF of these similar words in the second cluster can be attributed to the increase in caution related words in the first cluster.
(TIF)

**S2 Fig. Top 5 VADER words by TF-IDF before, during, and after Hurricane Dorian for each region.** For Florida and Alabama, the words in both clusters are similar across each time period. The Carolinas, on the other hand, show an increase in caution related words throughout (seen in cluster 1) and thus a decrease in the commonly said words (seen in cluster 2).
(TIF)

**S3 Fig. Bootstrapped LIWC positive word ratio (LWPR) around the hit date for each hurricane and location.** For each day, we calculated the 2.5th, 50th, and 97.5th percentile of the bootstrapped sentiments in both the original data and the null model. The bold line is the median of the observed data and the null model while the shaded areas represent the 95% confidence interval. Any day in which the confidence intervals does not intersect represents a statistically significant change in sentiment. In all locations with landfall, the positive word ratio had no significant change with the exception of Hurricane Irma 7 days after landfall.
(TIF)

**S4 Fig. Bootstrapped LIWC negative word ratio (LWNR) around the hit date for each hurricane and location.** For each day, we calculated the 2.5th, 50th, and 97.5th percentile of the bootstrapped LWNR in both the original data and the null model. The bold line is the median of the observed data and the null model while the shaded areas represent the 95% confidence interval. Any day in which the confidence intervals does not intersect represents a statistically significant change in the LWNR. Hurricanes Florence, Harvey, and Irma caused a significant increase in the LWNR starting at landfall while the LWNR in Dorian (Carolinas) dropped 2 days beforehand.
(TIF)

**S1 Data.**
(ZIP)

## Author Contributions

**Conceptualization:** Krishna Bathina, Marijn ten Thij, Johan Bollen.

**Data curation:** Krishna Bathina.

**Formal analysis:** Krishna Bathina, Marijn ten Thij, Johan Bollen.

**Funding acquisition:** Johan Bollen.

**Investigation:** Krishna Bathina.

**Methodology:** Krishna Bathina, Marijn ten Thij, Johan Bollen.

**Resources:** Johan Bollen.

**Visualization:** Krishna Bathina, Marijn ten Thij, Johan Bollen.

**Writing – original draft:** Krishna Bathina, Marijn ten Thij, Johan Bollen.

**Writing – review & editing:** Krishna Bathina, Marijn ten Thij, Johan Bollen.

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
