## [Decision Letter · Decision Letter 0]

2 Dec 2021

PONE-D-21-22890

Quantifying societal emotional resilience to natural disasters from geo-located social media content

PLOS ONE

Dear Dr. Bathina,

Thank you for submitting your manuscript to PLOS ONE. After careful consideration, we feel that it has merit but does not fully meet PLOS ONE’s publication criteria as it currently stands. Therefore, we invite you to submit a revised version of the manuscript that addresses the points raised during the review process.

Reviewer 1 & 2 offer suggestions for improving the manuscript that I find quite reasonable and manageable. Please address each of their comments & concerns in a revision.

We look forward to receiving your revised manuscript.

Kind regards,

Christopher M. Danforth

Academic Editor

PLOS ONE

Journal Requirements:

2. In your Methods section, please include additional information about your dataset and ensure that you have included a statement specifying whether the collection method complied with the terms and conditions for the website.

3. Please ensure that you refer to Figure 5 in your text as, if accepted, production will need this reference to link the reader to the figure.

Reviewers' comments:

Reviewer's Responses to Questions

**Comments to the Author**

1. Is the manuscript technically sound, and do the data support the conclusions?

Reviewer #1: Partly

Reviewer #2: Yes

Reviewer #3: Partly

2. Has the statistical analysis been performed appropriately and rigorously? 

Reviewer #1: Yes

Reviewer #2: Yes

Reviewer #3: Yes

3. Have the authors made all data underlying the findings in their manuscript fully available?

Reviewer #1: No

Reviewer #2: No

Reviewer #3: Yes

4. Is the manuscript presented in an intelligible fashion and written in standard English?

Reviewer #1: Yes

Reviewer #2: Yes

Reviewer #3: Yes

5. Review Comments to the Author

Reviewer #1: This paper provides an engaging analysis of the emotional dynamics of online speech in response to natural exogenous shocks. The sentiment time series is well done, with two sentiment analysis tools and bootstrapped estimates of confidence intervals. The analysis of sentiment dynamics is an interesting result in and of itself. However, the manuscript would benefit from strengthening the connection between the analysis, which is strong, and the claim that this is representing community emotional response.

In Table 1 the number of raw tweets collected for each storm is presented, but this could be misleading since retweets and non-English tweets are excluded, especially since the number of tweets for Dorian is already on the low side. I think it would be better to show the number of tweets actually used in the analysis here.

For Figure 2, it would be nice to have context with a map for exactly what communities are included within the bounding boxes. If the plot is made in python, something like the package "contextily" might help to make this easy.

I’d also like to know how the scale of the bounding box effects the results. Are the bounding boxes very tightly wrapped around the hurricane’s path of destruction, or is much of the surrounding area included? Wide enough that people who evacuated are still within the box? At least presenting the scale of the bounding boxes, would be helpful for readers.

Perhaps this is a topic for a separate paper, but seeing how the emotional content of tweets vary spatially would be very interesting. How concentrated are negative tweets are in areas with severe destruction vs surrounding communities? Does average tweet sentiment become more neutral further from the coast? In the abstract the authors state: "These effects may be distributed unequally, affecting some communities more profoundly and possibly over longer time periods than others", and a finer level of geographic analysis could shed some light here if we can measure this on using social media data, beyond a binary model of affected, non-affected.

I’d like to see the results of the exponential fits presented better and discussed more for the reader. What specifically are you interested in learning from these fits? If the timescale of emotional recovery is of interest, presenting the estimated exponential half-life in units of days, would allow readers to compare to other studies. Looking at reference 30 "Exploring the impact of a natural disaster on the health and psychological well-being of older adult" this study finds symptoms of stress from hurricanes takes 16 months to resolve, so what do we make of the reported exponential fits in this study? Two orders of magnitude longer relaxation time between individual resilience and community resilience. Expanding on what specific processes you think is being measured in this study vs some of the other studies cited would be valuable. Maybe Twitter data looks better for measuring some kinds of processes than others, and targeted survey’s will continue to be necessary for measuring others.

Some minor issues:

In the data section, the authors state data is collected between 2016 to 2018 but study Dorian is from 2019.

Should the clip art in Figure 1 show a tornado?

There are some latex errors, especially related to figures and citations which should be resolved before publication.

It looks like the authors forgot to include the link to the data repo with Tweet IDs.

Reviewer #2: The authors have demonstrated effectively an innovative and useful method to gain additional insights into the attitudes and emotions of residents before, during and after major hurricanes disrupt communities. The research objectives are presented clearly. The method is well-described and should be generalizable to furture storms. The figures are effective in helping the reader understand the analysis.

Finally, the discussion section of the paper includes a statement of the limitations of the study which is helpful.

That section could be improved by adding information concerning the following:

1) Adding some additional information about the "digital divide", the documented disparities in social-media use among various sub-groups within a community, would strengthen the discussion section of the manuscript. For example, are the older or less-affluent residents - those who would be more vulnerable during and after the storm - less likely to communicate via social media?

2) The discussion section would be improved by adding a brief consideration of how wide-spread electric power outages and cell-tower damages would limit Twitter communications following a major storm.

3) Including specific detail about the relative popularity of Twitter compared to other social media platforms - especially among younger residents would be helpful. (The 2018 Pew Social Media Fact Sheet) What percentage of the public use Twitter?

Reviewer #3: The paper presents a study looking at twitter's data during crises - authors focus on hurricanes data. The goal is to explore sentiment changes are the hurricanes are expected and eventually make landfalls. I have several concerns about this paper:

- the paper is very straightforward, authors use existing tools and public datasets for a simple analysis of sentiment, using vader and liwc.

- the dataset is small

- the tools are used as-is, with little to any insights on validity of either (vader and liwc)

- the analysis is a simple comparison of results

- there are writing issues and missing references.

Put simply, while the topic may be of interest, the paper lacks scientific depth and appears to be a simple exercise of data analytics (basic statistics) on public data. findings are hard to generalize and do not seem to reveal significant insights.

6. PLOS authors have the option to publish the peer review history of their article (what does this mean?). If published, this will include your full peer review and any attached files.

Reviewer #1: No

Reviewer #2: No

Reviewer #3: No

---

## [Author Response · Author response to Decision Letter 0]

4 Apr 2022

The reviewers made a positive evaluation of our manuscript and offered many constructive suggestions. We are truly grateful for their thoughtful feedback which was instrumental in finalizing our revisions. We believe fully addressed all reviewer comments and are hopeful our manuscript is now ready for publication. In this rebuttal, we first address the editor comments and proceed with addressing all comments on a line-by-line basis. 

Editor:

Have the authors made all data underlying the findings in their manuscript fully available?

Our Response: 

We have added a link to the github repository containing the data used in the analysis in the second paragraph of the ‘Data and Methods’ section.

A list of Tweet IDs for tweets used in this analysis can be found at https://github.com/kbathina/resilience-to-natural-disasters. All data from OSoMe and the Twitter API were collected in accordance with their respective terms and conditions.

Reviewer 1:

Comment 1

This paper provides an engaging analysis of the emotional dynamics of online speech in response to natural exogenous shocks. The sentiment time series is well done, with two sentiment analysis tools and bootstrapped estimates of confidence intervals. The analysis of sentiment dynamics is an interesting result in and of itself. However, the manuscript would benefit from strengthening the connection between the analysis, which is strong, and the claim that this is representing community emotional response.

Our Response:

We agree that a stronger connection is needed between the analysis and our claim of representing community emotional response. We have added a paragraph (paragraph 7) in the introduction citing previous research relevant to community emotional response. Below is the added text:

Our work also builds upon previous research in social media analysis and community response. Bollen et al. performed sentiment analysis on Tweets from 2008 and found that changes in sentiment correlated with major and notable events that year [Bollen 2011]. Bollen et al. and Yang et al. both studied Twitter data and found found sentiment to be correlated to stock market returns [Yang 2015, Bollen 2011]. Recently, there have also been many papers studying Twitter sentiment during the COVID-19 pandemic, specifically focusing on the the effects of lockdowns and restrictions [Zhou 2021,Sharma 2020,Dubey 2020,Boon 2020].

Comment 2

In Table 1 the number of raw tweets collected for each storm is presented, but this could be misleading since retweets and non-English tweets are excluded, especially since the number of tweets for Dorian is already on the low side. I think it would be better to show the number of tweets actually used in the analysis here.

Our Response:

The number of tweets shown in the table is the number used in the analysis. We acknowledge our confusing sentence and have made it more clear in the second paragraph of the ‘Data and Methods’ section. Below is the added text:

The number of analyzed Tweets for each hurricane and region are shown in Table 1.

Comment 3

For Figure 2, it would be nice to have context with a map for exactly what communities are included within the bounding boxes. If the plot is made in python, something like the package "contextily" might help to make this easy.

Our Response:

Thank you for the recommendation. We have updated the plot using the contextily package in order to highlight the communities that were affected.

Comment 4

I’d also like to know how the scale of the bounding box effects the results. Are the bounding boxes very tightly wrapped around the hurricane’s path of destruction, or is much of the surrounding area included? Wide enough that people who evacuated are still within the box? At least presenting the scale of the bounding boxes, would be helpful for readers.

Perhaps this is a topic for a separate paper, but seeing how the emotional content of tweets vary spatially would be very interesting. How concentrated are negative tweets are in areas with severe destruction vs surrounding communities? Does average tweet sentiment become more neutral further from the coast? In the abstract the authors state: "These effects may be distributed unequally, affecting some communities more profoundly and possibly over longer time periods than others", and a finer level of geographic analysis could shed some light here if we can measure this on using social media data, beyond a binary model of affected, non-affected.

Our Response:

This is a very interesting extension to our work. The effects of the bounding box can not only change the magnitude of the emotional response but also the timing in relation to the hurricane hit date. This idea is something we will consider when working on future projects.

Comment 5

I’d like to see the results of the exponential fits presented better and discussed more for the reader. What specifically are you interested in learning from these fits? If the timescale of emotional recovery is of interest, presenting the estimated exponential half-life in units of days, would allow readers to compare to other studies. Looking at reference 30 "Exploring the impact of a natural disaster on the health and psychological well-being of older adult" this study finds symptoms of stress from hurricanes takes 16 months to resolve, so what do we make of the reported exponential fits in this study? Two orders of magnitude longer relaxation time between individual resilience and community resilience. Expanding on what specific processes you think is being measured in this study vs some of the other studies cited would be valuable. Maybe Twitter data looks better for measuring some kinds of processes than others, and targeted survey’s will continue to be necessary for measuring others.

Our Response:

We agree that the timescale of emotional recovery is of primary interest and have added the half-ilfe in days in Table 2. We have also explored the differences between the resilience recovery time in our results versus reference 30. We have the below text in the 5th paragraph of the ‘Results’ section.

Krause et al. found it took 16 months for an individual to mentally recuperate from the stressors of natural disasters [Kruase 1987]. One theory to explain the differences in recovery time with our results is that two different types of thinking are being measured [Kahneman 2011]. The quick resilience can be explained by an immediate and innate relief (Type 1 thinking) immediately after the primary disaster of the hurricane. The longer resilience can be explained by a logical and more analytical thinking (Type 2 thinking) involved in fully realizing the long- and short term effects of the hurricane.

Comment 6

In the data section, the authors state data is collected between 2016 to 2018 but study Dorian is from 2019.

Our Response:

Thank you for catching that mistake. We have fixed the year to 2019.

Comment 7

Should the clip art in Figure 1 show a tornado?

Our Response:

We have replaced the image to better represent a hurricane.

Comment 8

There are some latex errors, especially related to figures and citations which should be resolved before publication.

Our Response:

We have gone through and fixed all of the errors

Comment 9

It looks like the authors forgot to include the link to the data repo with Tweet IDs.

Our Response:

We have added a link to the github repository containing the data used in the analysis in the second paragraph of the ‘Data and Methods’ section.

A list of Tweet IDs for tweets used in this analysis can be found at https://github.com/kbathina/resilience-to-natural-disasters. All data from OSoMe and the Twitter API were collected in accordance with their respective terms and conditions.

Reviewer 2:

Comment 1

Adding some additional information about the "digital divide", the documented disparities in social-media use among various sub-groups within a community, would strengthen the discussion section of the manuscript. For example, are the older or less-affluent residents - those who would be more vulnerable during and after the storm - less likely to communicate via social media?

Our Response:

We agree that discussing the digital divide is extremely relevant to this work. We have included a paragraph in the ‘Discussion’ (paragraph 3) detailing the divide, its effects during natural disasters, and cited some previous studies. The text is presented below.

However, we believe that internet and social media usage during natural disasters is more nuanced when taking into account the "Digital Divide" [Rogers 2001]. As mentioned above, the Twitter population is not a representative sample of the world. For example, young adults tend to use social media more often than older adults [Pew]. Kent and Capello showed that during the Horsethief Canyon Fire in September 2012, younger people had disproportionately more fire related Twitter posts [Kent 2013]. Xiao et al. showed that during Hurricane Sandy, the number of related tweets increased in mildly damaged places and decreased in more affected areas [Xiao 2015]. They also found that young, male, or more educated populations are more likely to have more tweets. When looking at income disparity and tweet frequency, they found a reversed U-shaped pattern and hypothesized that an increase in wealth can indicate better access to technology and information. But, after a certain point, the extremely wealthy are less motivated to use social media, potentially because they are less affected by the natural disaster. Furthermore, Wang et al found similar results, showing that communities with higher social vulnerabilities were underrepresented even before Hurricane Sandy made landfall, indicating less social media engagement and the potential to be left behind during disaster recovery due to low visibility [Wang 2019]. We acknowledge the Digital Divide as a limitation in our analysis as we are not able to account for these population differences.

Comment 2

The discussion section would be improved by adding a brief consideration of how wide-spread electric power outages and cell-tower damages would limit Twitter communications following a major storm.

Our Response:

We agree that the effects of power outages is a confounding factor when considering twitter communications. We have added the below text as a limitation in paragraph 3 of the ‘Discussion.’

Hurricanes and other extreme weather can lead to power outages, sometimes lasting for days. Consequently, social media and general internet usage in some users can decrease during those times, also leading to a reliance on more traditional sources of information such as TV and radio [Burger 2013, Jennex 2012].

Comment 3

Including specific detail about the relative popularity of Twitter compared to other social media platforms - especially among younger residents would be helpful. (The 2018 Pew Social Media Fact Sheet) What percentage of the public use Twitter?

Our Response:

We have included some details about social media prevalence from the Pew 2018 Social Media Fact Sheet, shown below. This is the first paragraph located in the ‘Data and methods’ section.

We use social media for this analysis because of its pivotal position in public discourse. According to the Pew Research Center, 84% of US adults aged 18-29 used social media in 2021. Among those older than 65 years, 45% report that they are social media users. Even though Facebook has the most users (69% of US adults), we specifically use Twitter due to its microblogging nature; it is most commonly used to discuss daily routine and reporting news [Java 2007] while still representing 23% of all social media users. This is particularly important for our analysis as we are interested in people's evolving reports of their personal state in the context of a hurricane.

---

## [Decision Letter · Decision Letter 1]

19 May 2022

Quantifying societal emotional resilience to natural disasters from geo-located social media content

PONE-D-21-22890R1

Dear Dr. Bathina,

We’re pleased to inform you that your manuscript has been judged scientifically suitable for publication and will be formally accepted for publication once it meets all outstanding technical requirements.

Kind regards,

Miquel Vall-llosera Camps

Senior Editor

PLOS ONE

Reviewers' comments:

Reviewer's Responses to Questions

**Comments to the Author**

1. If the authors have adequately addressed your comments raised in a previous round of review and you feel that this manuscript is now acceptable for publication, you may indicate that here to bypass the “Comments to the Author” section, enter your conflict of interest statement in the “Confidential to Editor” section, and submit your "Accept" recommendation.

Reviewer #1: All comments have been addressed

2. Is the manuscript technically sound, and do the data support the conclusions?

Reviewer #1: (No Response)

3. Has the statistical analysis been performed appropriately and rigorously? 

Reviewer #1: (No Response)

4. Have the authors made all data underlying the findings in their manuscript fully available?

Reviewer #1: (No Response)

5. Is the manuscript presented in an intelligible fashion and written in standard English?

Reviewer #1: (No Response)

6. Review Comments to the Author

Reviewer #1: (No Response)

7. PLOS authors have the option to publish the peer review history of their article (what does this mean?). If published, this will include your full peer review and any attached files.

Reviewer #1: No

---

## [Editor Report · Acceptance letter]

7 Jun 2022

PONE-D-21-22890R1 

Quantifying societal emotional resilience to natural disasters from geo-located social media content 

Dear Dr. Bathina:

I'm pleased to inform you that your manuscript has been deemed suitable for publication in PLOS ONE. Congratulations! Your manuscript is now with our production department. 

Kind regards, 

on behalf of

Dr. PLOS Manuscript Reassignment 

Staff Editor

PLOS ONE